# Using SMART Magnetic Fluids and Gels for Prevention and Destruction of Bacterial Biofilms

**DOI:** 10.3390/microorganisms11061515

**Published:** 2023-06-07

**Authors:** Jarosƚaw E. Król, Garth D. Ehrlich

**Affiliations:** 1Center for Surgical Infections and Biofilms, Center for Advanced Microbial Processing, Center for Genomic Sciences, Department of Microbiology and Immunology, Drexel University, Philadelphia, PA 19104, USA; ge33@drexel.edu; 2Department Head and Neck Surgery, Drexel University, Philadelphia, PA 19104, USA

**Keywords:** biofouling, *E. coli*, nanoparticles, magnetic field, urinary catheters, flow cells, air–liquid interface, submerged biofilms, laser surface texturing

## Abstract

Biofouling is a major problem in all natural and artificial settings where solid surfaces meet liquids in the presence of living microorganisms. Microbes attach to the surface and form a multidimensional slime that protects them from unfavorable environments. These structures, known as biofilms, are detrimental and very hard to remove. Here, we used SMART magnetic fluids [ferrofluids (FFs), magnetorheological fluids (MRFs), and ferrogels (FGs) containing iron oxide nano/microparticles] and magnetic fields to remove bacterial biofilms from culture tubes, glass slides, multiwell plates, flow cells, and catheters. We compared the ability of different SMART fluids to remove biofilms and found that commercially available, as well as homemade, FFs, MRFs, and FGs can successfully remove biofilm more efficiently than traditional mechanical methods, especially from textured surfaces. In tested conditions, SMARTFs reduced bacterial biofilms by five orders of magnitude. The ability to remove biofilm increased with the amount of magnetic particles; therefore, MRFs, FG, and homemade FFs with high amounts of iron oxide were the most efficient. We showed also that SMART fluid deposition can protect a surface from bacterial attachment and biofilm formation. Possible applications of these technologies are discussed.

## 1. Introduction

Biofilms are complex multicellular microbial communities characterized by cells attached to the substrate surface, to interfaces, and/or to each other and embedded in an extracellular polymeric matrix that they have produced [1,2,3]. Biofilms are extremely important as they represent the preferred form of bacterial life in natural environments and provide bacterial resistance to antibacterial substances [4,5], making them major threats for human health and industrial processes [1,6,7,8,9,10]. Many technologies have been developed to fight and remove bacterial biofilms, including new antibiofilm materials, antibiotics and antimicrobial peptides, phages, matrix-degrading enzymes, biometallohydrogels, disinfectants, electrochemical, photochemical, and mechanical methods [11,12,13,14,15,16,17]. In our group, we develop new technologies to grow and analyze bacterial biofilms in order to find efficient methods of biofilm destruction [18,19,20,21,22]. Previously, we used a magnetic pin technique (DOI:10.13140/RG.2.2.30323.99368) to study bacterial biofilms. After watching a magnetic window washer and SMART fluid (SMARTF) commercials on YouTube, we were intrigued about testing the ability of a combination of these two technologies to destroy bacterial biofilms. SMARTFs are fluids whose properties can be changed by applying an electrical or magnetic field [23,24]. Magnetic SMARTFs can be divided into magnetorheological fluids (MRFs) and ferrofluids (FFs) based on their magnetic particle sizes and concentrations. MRFs are different from FFs, which have smaller particle sizes at lower concentrations. MRF particles are primarily on the micrometer scale and are too dense for Brownian motion to keep them suspended (in the lower-density carrier fluid). FF particles are primarily nanoparticles that are suspended by Brownian motion and generally will not settle under normal conditions [24,25]. Typical SMARTFs contain magnetic particles [usually iron oxide (IO)], carriers, and surfactants and are characterized by viscosity and magnetic strength [25,26]. In SMARTFs, the carriers are most commonly based on water, mineral oil, hydrocarbon, kerosene oil, and chitosan with water, with kerosene being the most common [25,26]. Ferrogels (FGs), also called magnetorheological elastomers (MREs), use a nonmagnetic elastomeric carrier. FGs are considered more environmentally friendly than SMARTFs [25,27]. SMARTF and FG applications have expanded rapidly in the last decade. The major areas are mechanical engineering and automotive, military and defense, optics, aerospace, water treatment, cosmetics, and biomedical [26]. Among biomedical applications, the most interesting are prosthetics and exoskeletons, biosensors and bioimaging technologies, hyperthermia therapy, anticancer drug delivery, antimicrobial activity, and diagnostic microfluidic devices (lab-on-chip) [26,28,29]. The toxicity of ferrofluid is very low; animal experiments in which the permissible maximum doses of ferrofluid were either administered orally, intravenously, or intraperitoneally did not kill the mice [30]. However, earlier investigators had noted that animal subjects (male Sprague-Dawley rats and male athymic nude mice) that received ferrofluid in the supraphysiological dose range developed lethargy for 12 to 24 h, a resistance of food uptake, and persistent discoloration for about one week; the later change was attributed to the relatively enormous iron load [31]. So far, only one case report of ferrofluid-associated cutaneous dyschromia has been described, which was completely resolved after 2 weeks without any adverse sequelae [32]. The antimicrobial activity of IO and other nanoparticles is based on their ability to form reactive oxygen species, which kill bacterial and fungal cells [26,33,34,35]. Recently, IO nanoparticles and nanorobots have been used to destroy bacterial biofilms [36,37,38,39].

Here, for the first time, we tested the capacity of commercially available SMARTFs and FGs, as well as some homemade FFs, to destroy bacterial biofilms in various laboratory models.

## 2. Materials and Methods

**Bacterial strains:** *Escherichia coli* C and an *E. coli* K12 MG1655 *csrA* mutant were grown on a Luria-Bertani (LB) broth (Miller) in all experiments [21,22,40]. Overnight cultures were grown in a 37 °C shaker at 200 rpm.

**SMART fluids (**Table 1**):** FFs and MRFs included the following: FerroTec EFH1; EFH3 (hydrophobic general, FF); PBG200 (hydrophilic, polyethylene glycol (PEG)-based, nonionic surfactant); EMG605 (hydrophilic, water-based, cationic surfactant); EMG705 (hydrophilic, water-based, anionic surfactant); EMG900; EMG901; EMG905; EMG909; EMG911 (hydrophobic, light hydrocarbon oil) (FerroTec (USA) Corporation, Santa Clara, CA, USA); PARKER LORD MRF-122EG (LORD Corp. Cary, NC, USA); AMT-Dampro; AMT-Magnaflo; AMT-Rheotec; and AMT-Smartec from ARUS MR TECH PVT LTD (Chennai, India). The FG tested was Aliver Black Luster Mask (FM) (Al’Iver, Hialeah, FL, USA). Homemade FFs included Alpha Chemicals black iron oxide natural (Fe_3_O_4_) in kerosene (different concentrations, see Table 1); Total Home Hand Soap (TH) (CVS, Woonsocket, RI, USA); HIGHMARK Dishwashing Liquid (DL) (Office Depot, Inc., Boca Raton, FL, USA); and Seventh Generation Inc. Powered by Plants Dish Liquid (PP) (Burlington, VT, USA). To make the TH, DL, and PP FFs, 10 g of black iron oxide was added into the 30 mL total volume. The viscosity of the homemade fluids was measured using an NDJ-5S Viscosimeter with S1 or S4 probes at room temperature.


**SMARTFs surface adhesion:**


To measure the interactions (stickiness/tackiness) between the SMARTFs and surfaces, we applied one hundred microliters of FF to the edge of the surface (tubes, slides), placed it vertically, and measured the distance of droplet migration and the amount of the FF attached to the wall (although the latest assay is qualitative only).


**Biofilm growth:**


*Culture tubes*: Thirteen milliliters of LB Miller broth was placed in culture tubes (VWR 47729-58), inoculated with 10 µL of overnight culture, and grown at room temperature at 50 rpm. The medium was changed every 48 h or 72 h.

*Glass slide biofilms:* Sterile glass microscope slides were placed in 50 mL conical tubes (Falcon, Corning Inc., Corning, NY, USA) containing 25 mL of LB Miller broth inoculated with 25 µL of overnight culture. The tubes were shaken at room temperature at 50 rpm. The slides with biofilm were transferred to tubes with fresh 25 mL medium every 48 h.

*Twenty-four-well plates*: Two milliliters of inoculation mixture (50 mL medium and 0.5 mL overnight culture of bacterial strain) was dispensed into the wells of 24-well plates. Biofilms were grown for 48 h up to 96 h.

*Catheters*: Gentle Catch Intermittent Urinary Catheters (Cat. No. 501005) (Gentle Cath, Oklahoma City, OK, USA) were cut at the tip and connected to sterile tubing with a luer-lock syringe port. A sterile 0.2-µm filter was placed in the drainage port. In the initial inoculation (10 µL of overnight culture in 10 mL of LB Miller broth), the catheters were filled to ~80% volume and placed vertically at room temperature. The liquid was exchanged with a fresh LB Miller broth every 48 h.

*Flow cells*: Vertical flow cells (BBS, LLC) were inoculated with 50 mL of overnight culture diluted 100× with LB Miller broth for 2 h. Low flow (200 µL/min) was applied using a Cole-Palmer Masterflex L/X peristaltic pump (Antylia Scientific, Vernon Hills, IL, USA).

*Corrugated ABS slides*: ABS textured plastic sheets (Technology Island, Amazon, Vernon Hills, IL, USA) originally textured on one side, were cut to the size of microscope slides and subjected to laser surface texturing (Appendix A) using a LaserPackerPro2 engraver (Shenzhen, China) with the following parameters: engraving resolution 2k, w45 h25 mm, power 100%, depth 10% pass 1.


**Magnetic biofilm removal:**


*Commercially available magnets:* In this study, (1) an N52 neodymium magnet (diameter OD × inner diameter × thickness: 25 × 16—to match culture tube OD × 5 mm) with a peak magnetic strength of 487 mG (as measured by an ERICKHILL EMF Meter); (2) an N52 neodymium magnet (19.05 × 6.35—to match catheter OD × 6.35 mm; peak 401 mG); (3) an N52 neodymium cube magnet (1 cm^3^, peak 496 mG) for flat surfaces; and (4) an N52 neodymium magnet (length × width × height: 60 × 10 × 3 mm; peak 405 mG) were used for the biofilm growth inhibition experiments.

*Culture tubes:* Culture tubes with biofilm were treated as follows. The liquid medium was replaced with 20 mL of MilliQ water (Merck Millipore, Burlington, MA, USA). Three hundred microliters of SMARTF or 0.3 g of FM was applied. An N52 neodymium magnet (diameter × inner diameter × thickness: 25 × 16 × 5 mm) with a peak magnetic strength of 487 mG, as measured by an ERICKHILL EMF Meter, was placed on the tube and slid 12 times. Cell debris were removed and 15 mL of crystal violet (CV) solution (0.2% in 20% ethanol) was used to stain for biofilm for 30 min. The tubes were washed with tap water (4×) and dried. The CV was solubilized with 30% acetic acid in ethanol (16 mL) and its absorbance was measured at 570 nm (Tecan Infinite, M200Pro, Männedorf, Switzerland).

*Glass slides*: Glass slides were washed in 50 mL of sterile saline and placed in a petri dish with 15 mL of sterile saline. SMARTF (100 µL) or FM (0.15 g) was applied to the slides. An N52 neodymium cube magnet (1 cm^3^, peak 496 mG) was used at the bottom of the dish to drag the magnetic particles along the slide. The same method was used with the other side of the slide after inverting it. The slides were washed 3 times with 50 mL of sterile saline and stained with CV (as described above). For CFU counts, the glass slides were placed in 25 mL of sterile saline in 50-mL conical tubes; a combination of scraping (cell scraper Fisher Sci. 08-100-241, Fisher Scientific, Hampton, NH, USA) and vortexing (5 min, 1500 rpm, Spex Mini-G Homogenizer, Spex Sample Prep, Metuchen, NJ, USA) was used to detach the cells. Serial dilutions were plated on LB Miller agar plates and grown overnight at 37 °C; the results were calculated as CFU per milliliter.

*Bio-Tek II chamber slides*: Four chamber slides (Bio-Tek, Winooski, VT, USA) were used for microscopic analyses [41]. The slides were inoculated with 1.5 mL or 0.35 mL of refreshed (1:100) overnight cultures of individual bacterial strains grown in LB-Miller medium. The medium was changed every 48 h. Fifty microliters of SMARTF or 0.1 g of FM was added to remove biofilms. An N52 neodymium cube magnet (1 cm^3^, peak 496 mG) was used at the bottom of the slide to drag the magnetic particles along the slide. For microscopy, 600 µL of acridine orange solution (0.1%) was used for staining (30 min). The slides were sealed and analyzed with a Keyence BZ-X710 All-in-One Fluorescence microscope (Keyence, Osaka, Japan). The pictures were analyzed using the Trainable Weka Segmentation [42] plugin of ImageJ Figi 1.53q software (National Institutes of Health, Bethesda, MD, USA).

*Catheters:* One hundred microliters of SMARTF or 0.15 g of FM was injected into the catheters with 12-day-old biofilms through the drainage port. An N52 neodymium magnet (diameter × inner diameter × thickness: 19.05 × 6.35 × 6.35 mm; peak 401 mG) was used to move the magnetic particles along the catheter. Th results were collected by microscopy using an Olympus BX60 microscope (Tokyo, Japan) with a 2× lens. The pictures were analyzed by ImageJ software.

**Biofilm growth inhibition:** An N52 neodymium magnet (length × width × height: 60 × 10 × 3 mm; peak 405 mG) was placed between two sterile microscope glass slides and glued using Gorilla Hot Glue Sticks (Cincinnati, OH, USA) and a glue gun. The slides were placed in 50-mL Falcon tubes containing 25 mL of LB Miller broth inoculated with 25 µL of overnight culture. The tubes were shaken at room temperature at 50 rpm. The slides with biofilm were transferred to 25 mL of fresh medium every 48 h. After six days, the slides were removed, washed 3× in 50 mL of water, and stained with CV solution (0.2% in 20% ethanol) for 30 min. The slides were washed with tap water and dried. The results were collected by microscopy using an Olympus BX60 microscope with a 2× lens. The pictures were analyzed by ImageJ software. Blue pixels on the pictures were counted using the Canvas Pixel Color Counter (https://townsean.github.io/canvas-pixel-color-counter/, accessed on 5 June 2022). The results are presented as the ratio of blue/total number of pixels. For experiments with Bio-Tek II chamber slides; the slides were inoculated as described above. Fifty microliters of SMARTF was added to each well and the slides were placed on N52 neodymium magnets (length × width × height: 60 × 10 × 3 mm; peak 405 mG) in a 28 °C incubator. The slides were processed, stained with acridine orange, and analyzed as described above.

**Viability assay:** The effect of SMARTFs on *E. coli* cells was tested as follows. One hundred milliliters of LB Miller broth was inoculated with 1 mL of overnight culture and incubated to an OD_600_ = 0.6 at 37 °C (200 rpm). The culture was added to Eppendorf tubes with 1 µL, 10 µL, and 100 µL of SMARTF to 1 mL final volume. Tubes were mixed well and stored at room temperature for 2 h. Serial dilutions were plated on LB Miller agar and plates were incubated overnight at 37 °C. Colonies were counted and the resulting CFU per milliliter was analyzed using a *t*-test.

***G. mellonella* larvae toxicity studies.** Galleria larvae provide a fast and convenient means to assess in vivo toxicity and results show a strong correlation to those from mammalian systems [43]. We injected 3 µL of the SMART fluid in the last left pro-leg. Larvae were kept at room temperature for 3 days, counting the number of dead and alive larvae every 24 h. We used PBS as a control.

## 3. Results

### 3.1. Properties of SMART Fluids

The performance of SMARTFs can be significantly affected by their physical and chemical characteristics, including saturation magnetization and viscosity; these can be combined in many ways, resulting in SMARTFs being suitable for a diverse array of applications. In our experiments, we used seven groups of SMARTFs with different properties (Table 1). Group 1 consists of standard hydrophobic educational FFs (FerroTec EFH1 and EFH3). These commonly used FFs utilize a light hydrocarbon as a carrier and differ in viscosity and saturation magnetization (44 and 65 mT, respectively). Saturation magnetization depends, to some extent, on the amount of IO nanoparticles in the fluid [44]. Group 2 contains FFs made with water as a carrier (FerroTec PBG200, EMG605, and EMG705). These FFs are hydrophilic and show slight differences in viscosity (see Table 1); however, the major difference is their surfactant charge. PBG200 has a nonionic surfactant, while EMG605 and EMG705 have polar surfactants (cationic and anionic, respectively). The saturation magnetization in this case is the same (22 mT). The third group consists of FerroTec EMG900-type FFs. These FFs differ in viscosity and the amount of IO nanoparticles (Table 1); therefore, the saturation magnetization doubles for each consecutive FF from 11 mT for EMG911 to 99 mT for EMG900. The fourth group consists of MRFs, which have much higher viscosity and levels of IO (Table 1). Nevertheless, the MRFs are still very easy to handle (pipetting). Groups 5 and 6 consist of lab-made FFs. Group 5 is based on kerosene and differs in IO percentage from 1.69% for K1 (which is below the consistency in all commercial FFs) to 54% in the case of K7 (which is a value between commercial FFs and MRFs). Group 6 consists of the same amount of IO with different carriers. We used commercial household detergents, such as liquid hand soaps and dishwashing liquids with different viscosities (Table 1). The viscosity of Group 6 FFs was 10 to 30 times higher than the most viscous MRFs. The idea behind the use of detergents was to help kill bacterial cells removed from the biofilm. Unlike the commercial SMARTFs, all of our lab-made FFs had a tendency to precipitate and separate IO from the carrier. The greatest precipitation was observed in Group 5 with the kerosene carrier. The seventh group contains the FG, the Aliver Face Peeling Mask (FM) which is a cosmetic industry product that has been approved for use on skin and therefore should be safe for any kind of surface medical application. This facemask has a cream-like consistency and has the highest viscosity (Table 1).

### 3.2. Toxicity Effect of SMART Fluids on Bacteria and G. mallonella

SMARTFs have been used in medical applications and should have low levels of toxicity. However, it has been described before that IO nanoparticles have a bactericidal effect by means of generating reactive oxygen species [45]. As our SMARTFs used differently coated nanoparticles (anionic, cationic) as well as pure IO, we needed to test their toxicity. In addition, the composition of the carrier fluid could affect cell viability. In particular, the lab-made FFs were designed to have a bactericidal effect with surfactants, including sodium laureth sulfate, sodium lauroyl lactylate, and sodium lauryl sulfate, and killing agents, such as methylchloroisothiazolinone and methylisothiazolinone (https://www.chemicalbook.com/, accessed on 25 July 2022).

In our experiments, we added 0.1%, 1%, and 10% of SMARTF solution to exponentially-grown *E. coli* C cultures (OD_600_ = 0.6) and checked the cell viability after 2 h. Surprisingly, only in two commercial FFs (EMG605 and EMG911) did we notice a statistically significant reduction in bacterial cell number in a dose-dependent manner (Figure 1). A similar pattern was observed in the case of MRFs, except for MRF-120EG. The difference between EMG605 and EMG705 is the character of the surfactants used. In EMG605, nanoparticles are covered by a cationic surfactant, while in EMG705, an anionic surfactant is used. It has been shown that cationic surfactants, particularly when applied at alkaline to neutral pH, show a high affinity for the interfaces of all microbial classes due to the negative charge of the microbial interfaces [46].

Smart fluids are considered to be nontoxic to higher organisms [30]. However, we tested our FFs to see their effect on *G. mallonella* larvae survival. Our experiments showed that the survival rate for most of the tested FFs was above 90%. Only some AMT SMARTFs showed a lower survival rate (AMT-Magnaflo—66%, AMT-Dampro—78%, AMT-Smartec—70%); although, the death was observed only within the first 24 h, suggesting that some other factors might be involved.

### 3.3. Surface Adhesion

All FFs have a tendency to stick and stain surfaces. Surface adhesion depends on the viscosity and surface tension (PHYSICAL REVIEW E 70036311 (2004)). As we used different materials, we first checked how the FFs interact with a smooth surface, both dry and with a solvent (water). All tested FFs stuck to the glass test tubes and plastics. We observed that the most viscous fluid TH and gel FM covered the surface and did not even make it to the bottom of the tube. The adhesion depended on the viscosity.

### 3.4. Removing Biofilm from Glass Tubes

As biofilms are often observed in pipes and pipelines, we used a culture tube to simulate biofilm removal from a cylindrical surface. We grew biofilms for 8 days and tested the efficiency of SMARTFs to remove the attached biofilm. We used a ring magnet with a peak strength of 487 mG, which caused the formation of an FF ring inside the tube (Appendix A). The first experiment showed that the water-based FFs from Group 2 could not be used. The FFs added to the water solution not only became diluted but also covered the entire internal surface protecting the biofilm from being removed. All hydrophobic SMARTFs worked as predicted, forming a ring inside the tube. A magnet was moved over the outside surface of the tube 12 times; however, even a single pass removed the visible bacterial biofilm ring from the surface (Appendix A). Analyzing the data of Groups 1 and 3, we noticed that the more viscous FFs with a higher content of magnetic nanoparticles worked better than their less viscous and low magnetic strength counterparts; i.e., EHF3 (12 mPA∙s, 12%) worked better than EHF1 (6 mPA∙s, 7.8%) (Figure 2). This trend was very clear, especially in Group 3 (EMG900 series), as both EMG900 and EMG911 were statistically different from other members of this group (Figure 2). However, a comparison between these two groups showed that the efficiency might also depend on the carrier. EHF1 (6 mPA∙s, 7.8%) should be comparable with EMG905 (5 mPA∙s, 7.8%), while EHF3 (12 mPA∙s, 12%) should give results between those of EMG900 and EMG901 (60 mPA∙s, 17.7%, and 10 mPA∙s, 11.8%, respectively). The results showed that Group 1 (EHF) works better than Group 3 (EMG). The difference is the carrier—light hydrocarbon for EHF and light hydrocarbon oil for EMG.

To test this hypothesis, we decided to make our own FFs using different carriers with the same concentration of IO nanoparticles. The amount of iron oxide placed in these FFs was between all commercially available FFs and MRFs (2–17.7% < 27% < 72–84%). In Group 5, we used kerosene (K5) and in Group 6, we used different household detergents (TH, DL, PP). These FFs had different viscosities, from 0 mPA∙s for K5 to 3162 mPA∙s, 5010 mPA∙s, and 6635 mPA∙s for PP, DL, and TH, respectively. We assumed that liquids with a higher viscosity should work more efficiently. The data confirmed this to some extent, as the difference between PP (10.2% of remaining biofilm; lowest 3162 mPA∙s) and TH (6.9%; highest 6635 mPA∙s) was statistically significant (*p* < 0.05, *t*-test). Unfortunately, the DL FF, which had an intermediate viscosity, did not follow the pattern (13.6% of remaining biofilm). Interestingly, the kerosene FF K5 with no measurable viscosity showed the highest activity in biofilm removal (99.9%) of all of the tested FFs (Figure 2), which showed that the simplest carrier and IO particles are enough for biofilm removal. Nevertheless, K5 FF has more than twice as many magnetic nanoparticles as the highest commercial FF (EMG900, 17.7%). To test the effect of magnetic nanoparticle concentration, we expanded Group 5 to include seven FFs (K1 to K7) with 1.69% to 54% of IO (Table 1) and tested their ability to remove biofilm. As expected, the biofilm removal increased with the IO concentration (Figure 3). For K1, with the lowest amount of iron oxide (1.69%), as much as 8.3% of the biofilm remained on the surface; meanwhile, after K6 and K7 treatment, the surface was cleaner than the control (negative results). All Group 5 kerosene series FFs showed enhanced biofilm removal compared to any of the commercial FFs (Figure 2).

As the percentage of magnetic particles affected biofilm removal, we tested the two last groups of SMARTFs: Group 4 consisting of commercial MRFs and Group 7 containing the FG (Aliver Magnetic Face Peeling Mask). MRFs contain a high percentage of magnetic particles (from 72% to 84%) with relatively low viscosity (43 to 240 mPa∙s) and are easy to handle. All MRFs removed the biofilm very efficiently (Figure 2), leaving a clean surface (negative values). Similarly, the FM removed all visible biofilm from the glass; however, the very high viscosity impacted the washing process, with some pieces of the FM being attached to the bottom of the tube. These remnants contained some biofilm cells and therefore gave a weak positive signal after staining with CV (0.4% of the remaining biofilm).

### 3.5. Removing Biofilm from the Flat Glass Slides

We used two methods of biofilm growth on the microscope slides: microscope slides in batch culture and Bio-Tek II chamber slides [41]. Microscope slides being used in a batch culture biofilm reactor is one of the methods allowing for biofilm formation on the air–liquid interface and is routinely used in our lab [18,19,20,22]. The chamber slide system has been used to analyze submerged biofilms attached to the bottom of wells [41]. In both systems, exchanging medium allows for indefinite biofilm growth and its microscopic analyses.

To analyze the efficiency of biofilm removal by SMARTFs, we used two standard methods: bacterial colony forming units (CFU) (Appendix A) and microscopic picture analyses. First, we removed biofilm from the glass slides (Figure 4A) and counted the remaining bacterial cells (Figure 4B). We noticed that all SMARTFs can remove biofilm; however, the efficiency of the commercial FF EHF3 is much lower (only a 2 log_10_ difference) than the homemade K5, MRFs, and FM (4 to 5 log_10_ difference). In particular, the AMT-Dampro was very efficient and removed 99.998% of the bacterial cells (Figure 4B).

Similar results were observed when analyzing microscopic pictures (Figure 5). Nontreated samples contained a robust biofilm (Figure 5A) while slides cleaned with the SMARTFs showed a few dozen bacterial cells, in the case of EHF3, K5, MRF-122EG, AMG-Smartec, and AMG-Rheotec (Figure 5B–D,G–I), to a few bacterial cells, for AMG-Dampro and AMG-Magnaflo (Figure 5E,F). We noticed that in the case of AMG-Dampro and AMG-Magnaflo, the slides remained covered with the liquid carrier and the bacterial cells were embedded in the oily matrix. In the case of EHF3 and K5, the remaining bacterial cells were encapsulated in water bubbles (vesicles) surrounded by the oily carrier (Figure 5B,C) or were covering the IO particles (Appendix A).

### 3.6. Biofilm Removal from Microtiter Plates

The most popular method for the growth and analysis of bacterial biofilms uses microtiter (96-, 48-, 24-well) plates [47,48]. We used 24-well plates as they offer a larger surface at the bottom of the plate. In our first *E. coli* C biofilm removal experiments using the crystal violet (CV) staining method, we noticed that the efficiency was not good for any of the tested SMARTFs (26.23% ± 11.29 of remaining biofilm). *E. coli* C is a very good biofilm former but in standard conditions (LB, 37 °C) still forms biofilm predominantly at the air–liquid interface and therefore cannot be removed by any SMARTF. In our search for a good, submerged biofilm former, we used an *E*. *coli* K12 *csrA* mutant grown at 28 °C in LB Miller [21,40]. The biofilm was stained with CV and analyzed by light microscopy (Appendix A). Surface coverage was analyzed as described before [42]. The results confirmed that the biofilms were removed with high efficiency (<96.4% clean surface) by K5, AMT-Dampro, AMT-Magnaflo, AMT-Smartec, AMT-Rheotec, and FM (Figure 6C,E–J). In the case of EHF3, all the structured biofilm was removed; although, more of the surface (69.1%) was still covered by a single layer of bacterial cells (Figure 6B,J). MRF-122EG removed the biofilm from 89.3% of the surface; although, most of the bacterial cells left were embedded in the MRF matrix (Figure 6D,J).

### 3.7. Removing Biofilm from Urinary Tract Catheters and Flow Cells

Catheter-acquired urinary tract infections (CAUTI) are one of the most common healthcare-acquired infections [49]. We used a catheter biofilm model to determine if the biofilm can be easily removed. The Gentle Catch Intermittent Urinary Catheter was used due to its relative transparency providing for direct observation. For this experiment, we used EHF3 and FM. FM was chosen because, as a skincare product, it might be applied directly to and have contact with human tissues. In both cases, the SMARTFs removed the biofilm (Figure 7); however, EHF3 covered and stained the catheter so strongly that macroscopic observations were highly reduced.

Similar experiments with the vertical flow cell showed the ability of FF and FM to remove biofilm and, in the case of FF, to fill all the gaps and cracks in the flow cell (Figure 8) (Appendix A).

### 3.8. Comparison of SMART Fluids to Other Methods of Removing Biofilms from Flat and Corrugated Surfaces

While studying the biofilm removal from the glass slides, we noticed that, unlike the magnetic cleaning, vortexing slides for 5 min at 1500 rpm alone was not enough to clean the slides. Therefore, we compared the efficiency of biofilm removal by FFs, FM, and standard methods, including scraping with a cell scraper and vortexing (1500 rpm, 5 min, Spex MiniG homogenizer). With the CV staining method, we showed that all protocols removed about 90% of the biofilm, between 89.75% and 94.15% for EFH3 and FM, respectively, with the FM showing the highest efficiency (between 91.5% and 97%) (Figure 9A). We noticed that standard deviations for the scraping, FM, and EFH1 were much higher than in the mechanical shaking method, which is an operator-independent method.

Flat and open surfaces, such as microscope slides, are easily accessible and relatively easy to clean attached biofilm from. Based on their mode of action, we hypothesized that FFs and FGs would remove biofilm more efficiently on a corrugated/textured surface (Appendix A). The results showed that the application of magnetic particles on a textured surface was much more efficient than mechanical scraping with the cell scraper (Figure 9B). However, we also noticed that hydrophobic FFs and the FM stuck to the hydrophobic ABS surface and were not easily controllable by a magnet.

### 3.9. Biofilm Formation Inhibition by SMART Fluids

Our experiments with catheters and flow cells revealed one more feature of SMARTF treatment. We noticed that both the catheters and flow cells after treatment showed a much lower efficiency of biofilm formation. Increased adhesion of FF to the surface raised a question about the possibility of surface protection from bacterial attachment and biofilm formation. To test this hypothesis, we placed a magnet between two glass slides, applied EFH1 on one side, and allowed the biofilm to grow for 10 days (Figure 10). The results showed that the area where FF was kept in place by the magnetic field was free of bacteria (Figure 10B). We also noticed that the surface not directly covered by the FF contained much less biofilm than the surface not exposed to the FF (Figure 10B). Quantitative picture analysis showed that the amount of biofilm on the treated side was almost six times lower than the not-treated control (15.38% and 91.42%, respectively). The presence of FF on one side of the slide did not affect bacterial growth or biofilm formation on the other side (Figure 10B). Similar results were obtained using the chamber slides (Figure 10C). We noticed that the places covered by the SMARTFs were cleaned of bacteria and the surrounding areas were also biofilm-free (Figure 10C). It was observed that the slide surfaces in all chambers, except control and K5, become extremely hydrophobic. This might explain the reduced bacterial attachment; however, the detailed mechanism of that process needs further investigation.

## 4. Discussion

Bacterial biofilms are the most common form of bacteria. Their complicated structures, the presence of extracellular matrices, and the presence of the bacteria in different physiological states (i.e., metabolically dormant persister cells) make biofilms extremely difficult to treat and eradicate. As standard antimicrobial treatments designed for planktonic bacteria fail when used for biofilms, new technologies must be developed. Among these new methods are those based on metal oxide nanoparticles [35,50,51,52,53]. IO nanoparticles are widely used in biomedical research because of their unique properties, good biocompatibility, low cytotoxicity, and straightforward synthesis. IO particles are also magnetic, which makes them easy to direct using a magnetic field [54,55]. IO particles have been used for decades to make magnetic fluids [56]. These fluids are generally called “SMART fluids” as they can change their properties in response to external triggers, such as electric or magnetic fields. SMARTFs differ in their “consistency” and can be divided into FFs (low-viscosity liquids, low iron oxide concentrations), MRFs (higher-viscosity liquids, high iron oxide concentrations), and FGs or MREs (high-viscosity gels or elastomers, high iron oxide or iron filing concentrations). Typical SMARTFs contain magnetic particles, a carrier, and soapy surfactants that prevent magnetic particles from clumping together and precipitating from the suspension. SMARTFs can be made at home, but they are commercially available and relatively cheap to purchase. In this work, we evaluated the ability of commercially available and homemade SMARTFs to detach bacterial biofilms under different laboratory settings. In search of a suitable FG, we came up with an Aliver Black Luster Mask (FM), which is a face-peeling cosmetics mask. The FM had a nice creamy consistency and strong magnetic properties. We also used some homemade FFs based on IO powder and different carriers (kerosene and some household detergents). The purpose of these detergents was to increase the killing of detached bacteria. In addition, the surfactants in these detergents were able to reduce the clumping and precipitation of iron oxide from the suspension. The only FFs where we noticed rapid precipitation were kerosene-based FFs (Group 5). Keeping the IO in suspension before each sampling was a challenge. Most of the SMARTFs were easy to handle and pipette; although, they always covered the surface and stained everything they encountered [32].

In our experiments, first, we tested the bactericidal effect of the SMARTFs. In the 2-h viability assay, only two FFs and four out of five MRFs showed a reduction in the bacterial population. Based on the properties of the bactericidal FFs, we can conclude that the toxic effect depends on the surfactant used in the FF. Unlike others, in EMG605, magnetic particles are covered with a cationic surfactant. Cationic surfactants, particularly when applied at alkaline to neutral pH (as with standard growth media), have shown a high affinity for the interfaces of all microbial classes; this is due to the negative charge of the microbial interfaces [46]. Therefore, we conclude that the bactericidal effect of EMG605 depends on the surfactant. In the EMG900 group, the only difference between the FFs is the IO nanoparticle concentrations; although, we do not know why only the EMG911 with the lowest IO concentration (2%) killed bacterial cells. Remarkably, none of the Group 6 homemade FFs with a combination of surfactants and antimicrobial agents affected bacterial survival (Figure 2). As these FFs contain 27% of IO, we conclude that the IO concentration alone cannot reduce bacterial growth. To support this conclusion, we have the data for the MRFs, where the concentration of magnetic microparticles in MRF-122EG was as high as 72% and still did not show any significant reduction in bacterial cell numbers. We do not know the exact composition of MRF-122EG and the AMG MRFs; however, all the AMG MRFs showed a bactericidal effect. The lack of bactericidal effect of the FFs was surprising, as many showed that IO nanoparticles kill bacterial cells at a MIC similar to that of standard antibiotics (30 µg/mL) [57,58]. However, others have reported much higher concentrations ranging from 560–800 µg/mL to 3 mg/mL [59,60] or no effect at all [61]. Darvish and colleagues pointed out the significance of the IO coating in the bactericidal properties of IO nanoparticles against *E. coli* and *S. aureus* [61]. Unfortunately, simply using the common household detergents as the carriers did not substitute for IO coating and did not affect bacterial survival.

Among our FFs, we distinguished Group 2 because of the water-based carrier liquid. Water-based FFs are described as more biocompatible and have been used, for instance, as contrast agents in magnetic resonance imaging and other biological applications [62,63,64]. We tested all of the SMARTFs for their surface adhesion and all of them adhered to the surface. However, after adding water to the culture tubes, all fluids, except those in Group 2, formed micelles and precipitated. Group 2 FFs resuspended in the water solution and stained the entire surface, including the biofilm, with a black/brownish color. The stained surface interacted with the CV solution and the distaining solution, making it impossible to quantify the biofilm. When the magnetic ring was applied, these FFs did not form any visible rings inside the tube. These two features made us decide to exclude Group 2 FFs from further experiments.

Some magnetic nanoparticles have been used before to destroy bacterial biofilms [36,38,52,65,66,67,68]. In these reports, the magnetic nanoparticles were designed and synthesized. Here, we decided to test the commercially available SMARTFs. Biofilm definitions are very broad and the research methodology lacks the necessary standards. Each method of biofilm growth and quantification has its limitations [69,70]. The most commonly used microtiter plate [47], Calgary device plate [71], and CDC bioreactor [72] methods have only limited or no use in biofilm destruction by magnetic nanoparticles. The last two methods have limited accessibility for standard magnets, while microtiter plates can be used but only with microscopy techniques for biofilm quantification. These limitations are due to the fact that most of the bacterial strains form biofilms at the liquid–air interface [19,73,74]. Therefore, it is possible to use and remove bacterial biofilm from the bottom of the well but not feasible to remove the bacteria attached to the walls. Here, we mainly used two simple methods—test tube and glass slide biofilms with standard CV staining or CFU counting, respectively. The test tube biofilm is easy to grow, allows changing the medium for prolonged growth, and is very easy to quantify with the standard CV staining method. Additionally, using ring magnets, either neodymium or electromagnets, allows the magnetic nanoparticles to form a ring inside the tube, which can be driven up and down along the wall simply by moving the magnet (Appendix A). A similar technique might be used in any round device, such as tubing and catheters, or flat device with limited access, such as flow cells.

Using our testing protocols, we found that all SMARTFs had the ability to destroy bacterial biofilms; however, they worked at different efficiencies (81.75% and 49.44% of removed biofilm for EHF3 and EMG911, respectively). Our homemade FFs were much more efficient and removed between 86.4% (DL) and 99.9% (K5) of the biofilm. We showed that the efficiency of biofilm removal depended on the percentage of magnetic particles in the FF. As a matter of fact, simply increasing the volume of the FF applied also increased the efficiency of biofilm removal. All MRFs used in our experiments removed biofilm very efficiently. In this case, CV assays showed reduced sensitivity (negative results; i.e., tubes were clearer than the control after MRF treatment) over CFU counting. Our data showed that the amount of biofilm can be reduced by 4–5 log_10_ and was consistent or even better than the previously published results [36]. As shown before, the efficiency of biofilm removal depended on the magnetic field rotations and movements [36,38]. We noticed that even a single pass of the magnet along the tube removed visible biofilm; although, the effect of a number of passes on the removal efficiency was not tested. We agree with others that the mechanism of SMARTF action on biofilm is strictly mechanical. Magnetic nanoparticles dragged by a magnetic field damage the biofilm matrix and cause bacteria detachment [36,66].

As many methods of biofilm removal have been described [11,12,13,14,15], we wanted to compare the efficiency of SMARTFs with the basic methods of scraping and shaking; we found that even the FFs (EHF3) performed at the same range, while FM was statistically more efficient than all tested methods. These differences were even more significant when we used a corrugated surface. In this case, bacteria can colonize rough surfaces [18] and hide in cracks and crevices from the scraper; SMARTFs with nano- or micro-particles can easily penetrate these spaces and efficiently remove bacteria.

Developing new antimicrobial materials and surfaces is a promising alternative to drug development [75]. Metal oxide coatings have been used to increase food and everyday safety [75,76]. As we showed here, SMARTFs can be used to protect surfaces from bacterial attachment and biofilm formation. In our experiment, we showed that the SMARTFs can protect the surface directly via mechanical cover but can also reduce the presence of attached bacteria in the surrounding areas. These effects might be induced by the diffusion of hydrophobic carrier liquid along the surface, as we noticed that after the treatment, with all but K5 SMARTFs, the glass surfaces became hydrophobic. The mechanism of this process needs to be studied further.

## 5. Conclusions

The relevance of magnetic nano- and micro-particles has increased over the last decade, as they have revolutionized many areas, including medicine (e.g., cancer theragnostic, biosensing, catalysis), agriculture, and the environment [29,77]. Here, we used flow cells and urinary catheters to prove the concept of using SMARTFs for biofilm removal. In all cases—flow cells (which can mimic water (liquid) barrels or tanks) and urinary catheters (tubing and pipes)—we showed that SMARTFs can mechanically remove internally attached bacterial cells and reduce bacterial load by four to five orders of magnitude. Biofilm removal efficiency is much higher than the standard scraping and vortexing methods, especially for biofilm grown on rough surfaces. SMARTFs can be also used to cover the surface and protect it from bacterial attachment. Many of the SMARTFs have been proven to be nontoxic [30]. Here we confirmed that tested SMARTFs have no or limited toxicity, even when injected into *G. mallonella* larvae. Therefore we think that the application of this technology to remove bacterial biofilms looks very promising. Combining magnetic particles with any antimicrobial ligands should even increase the efficiency of removing and killing bacterial cells [35,52,67].

## Figures and Tables

**Figure 1 microorganisms-11-01515-f001:**
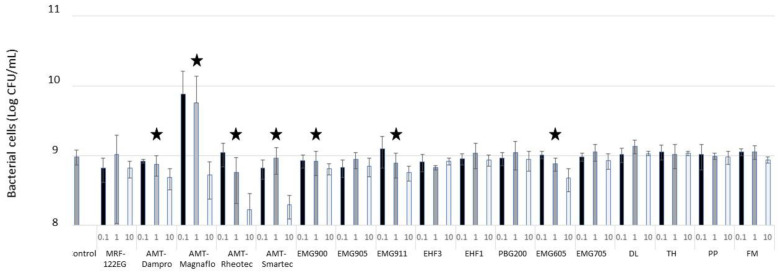
Effect of SMART ferrofluids on bacterial viability. SMARTF abbreviations are in Table 1. SMART fluids at various concentrations (0.1%, 1%, and 10%) were added to the exponentially-grown *E. coli* C cultures (OD600 = 0.6) and the cell density (CFU/mL) was measured after 2 h. Star symbols mark the SMART fluids under which the cell reductions showed a dose-dependent manner. Compared to the control (i.e., no exposure to SMART fluids), the star-labeled SMART fluids all showed significant cell reduction (*p* < 0.05, *t*-test) at a concentration of 10%.

**Figure 2 microorganisms-11-01515-f002:**
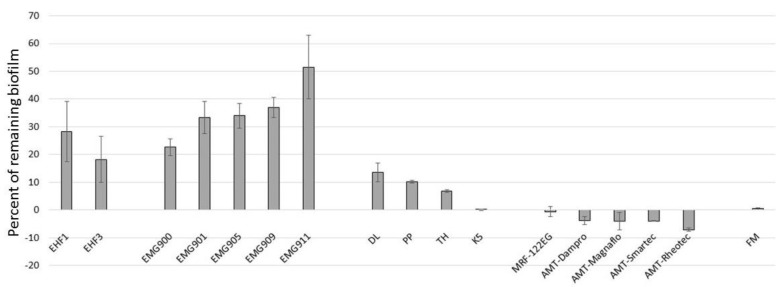
Efficiency of biofilm removal by SMART fluids in test tube experiment. The remaining cell number (CFU/mL) was measured and compared to the control (i.e., no exposure to the SMARTFs). The resulting percentage of biofilm on each condition was shown. SMARTF abbreviations and details are in Table 1. The “negative” remaining biofilm under the MRFs was because the MRFs removed the biofilm so thoroughly that the surface was much cleaner (i.e., contained less biofilm) than in the “no treatment” case.

**Figure 3 microorganisms-11-01515-f003:**
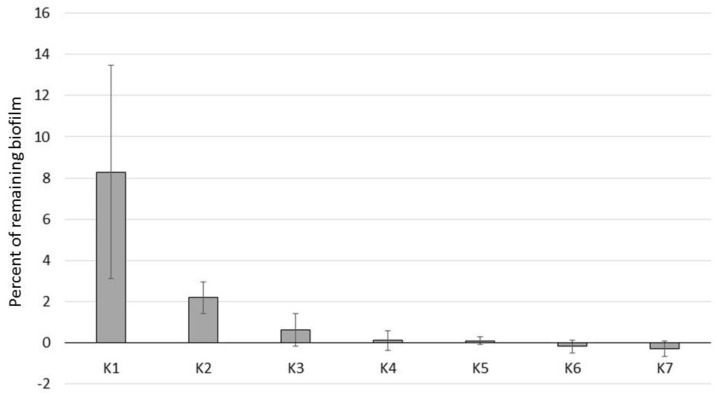
Effect of iron oxide concentration on biofilm removal. Iron oxide FFs at various concentrations (increasing from K1 to K7) were added to a biofilm of *E. coli* C grown on a culture tube for 8 days. The remaining cell number (CFU/mL) was measured and compared to the control (i.e., no exposure). The resulting percentage of biofilm on each condition was shown. See Table 1 for more details about the SMART fluids.

**Figure 4 microorganisms-11-01515-f004:**
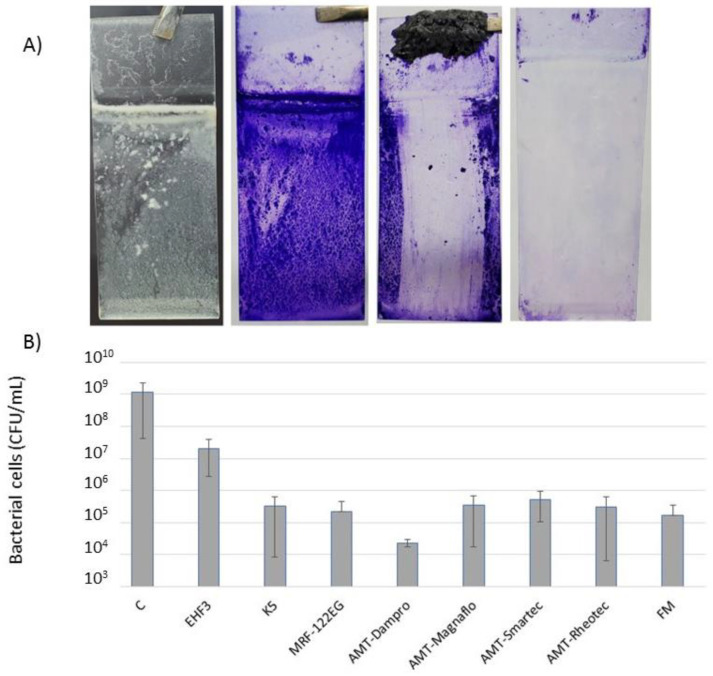
Biofilm removal from microscope slides. (**A**) Stages of biofilm removal treatment. From the left: slide biofilm, CV stained for better visualization, FM sliding, cleaned slide. (**B**) Efficiency of biofilm removal with SMART fluids presented as the number of remaining cells in CFU/mL. C—control without SMARTF treatment. See Appendix A for experiment details.

**Figure 5 microorganisms-11-01515-f005:**
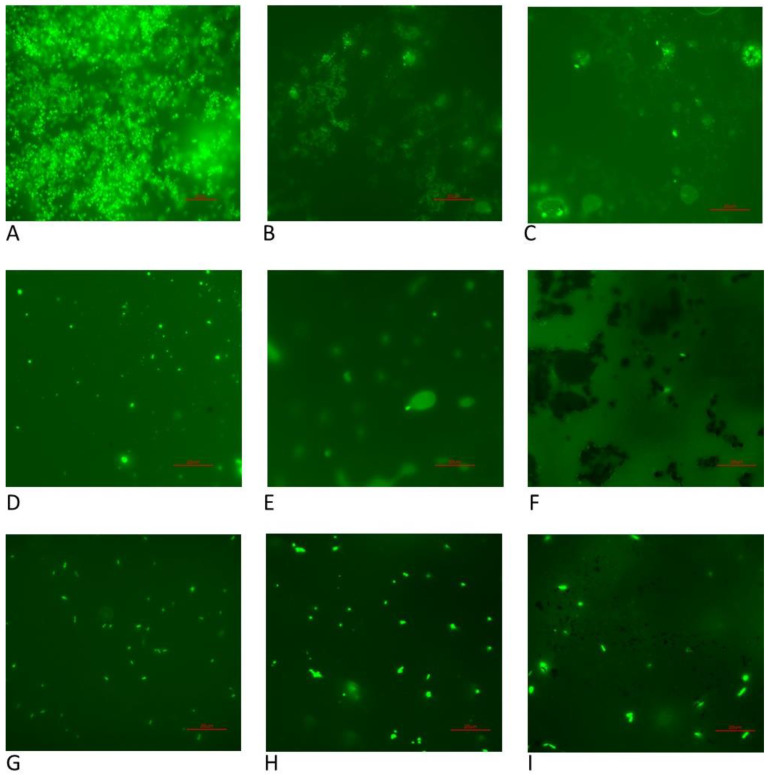
Microscopic analyses of *E. coli* slides. (**A**) Control, (**B**) EHF3, (**C**) K5, (**D**) MRF-122EG, (**E**) AMT-Dampro, (**F**) AMT-Magnaflo, (**G**) AMT-Smartec, (**H**) AMT-Rheotec, (**I**) FM. Red bar scales represent 20 µm.

**Figure 6 microorganisms-11-01515-f006:**
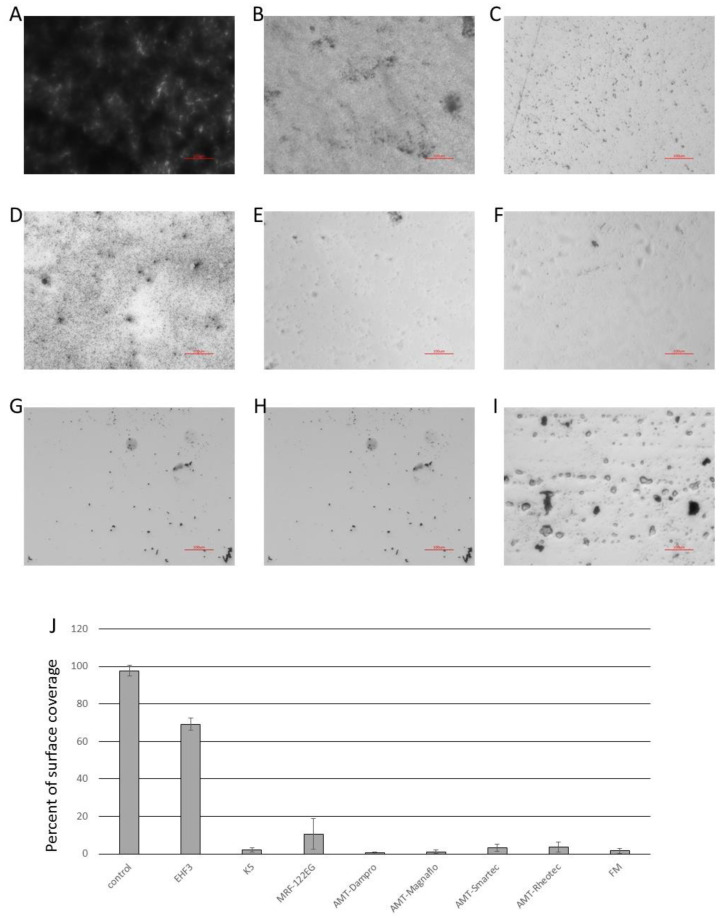
Microscope analyses of removal of submerged biofilm in a 24-well plate. (**A**) Control, (**B**) EHF3, (**C**) K5, (**D**) MRF-122EG, (**E**) AMT-Dampro, (**F**) AMT-Magnaflo, (**G**) AMT-Smartec, (**H**) AMT-Rheotec, (**I**) FM, (**J**) quantitative analysis of microscope pictures. Mag. 20×, Red bar scales represent 100 µm.

**Figure 7 microorganisms-11-01515-f007:**
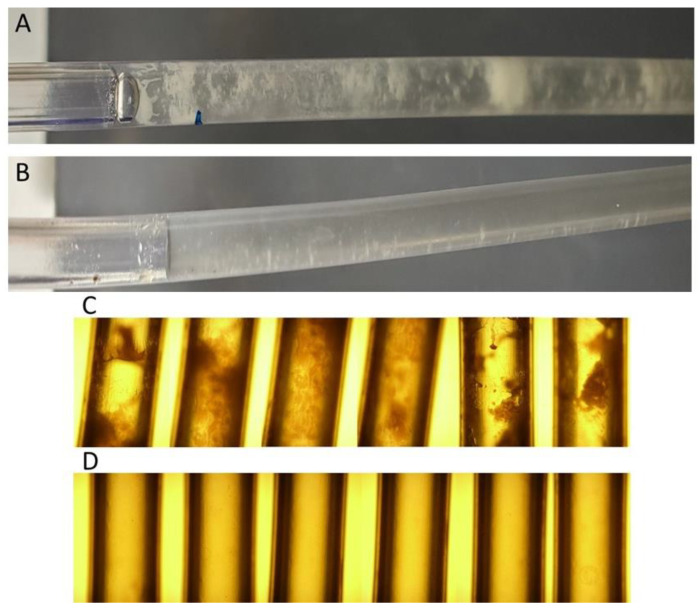
Twelve-day-old *E. coli* C biofilm in a urinary catheter before (**A**,**C**) and after (**B**,**D**) treatment with FM.

**Figure 8 microorganisms-11-01515-f008:**
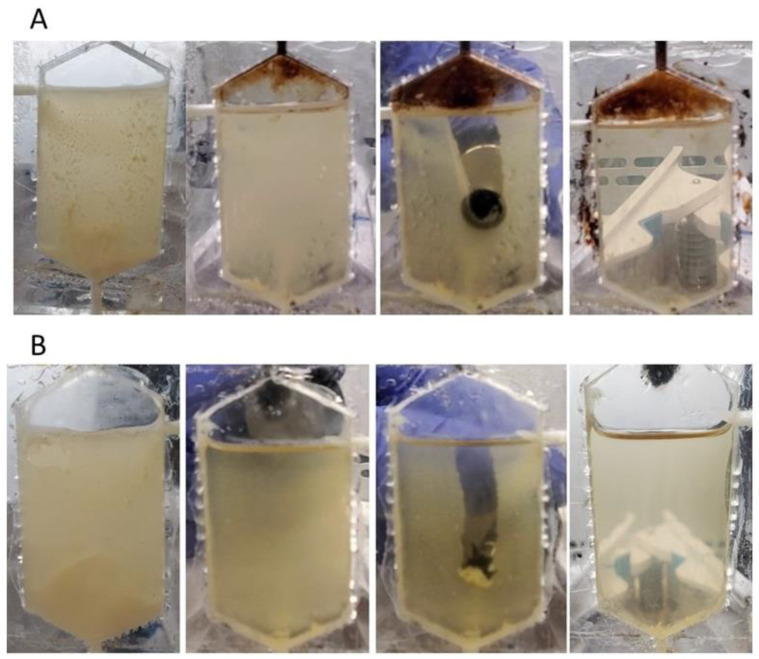
Removing biofilm from vertical flow cells using EHF3 (**A**) and FM (**B**). From the left: *E. coli* C biofilm on both sides of the flow cell; flow cell with one wall cleaned; removing biofilm from the second wall; flow cell with both sides cleaned.

**Figure 9 microorganisms-11-01515-f009:**
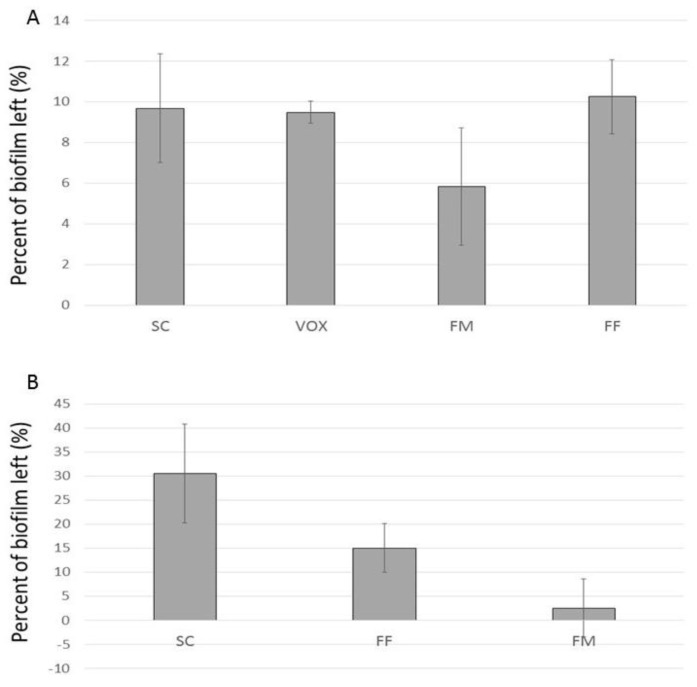
Efficiency of biofilm removal from glass slide (**A**) and ABS textured plastic (**B**) by cell scraping (SC), vortexing (VOX), FM, and EHF3 (FF).

**Figure 10 microorganisms-11-01515-f010:**
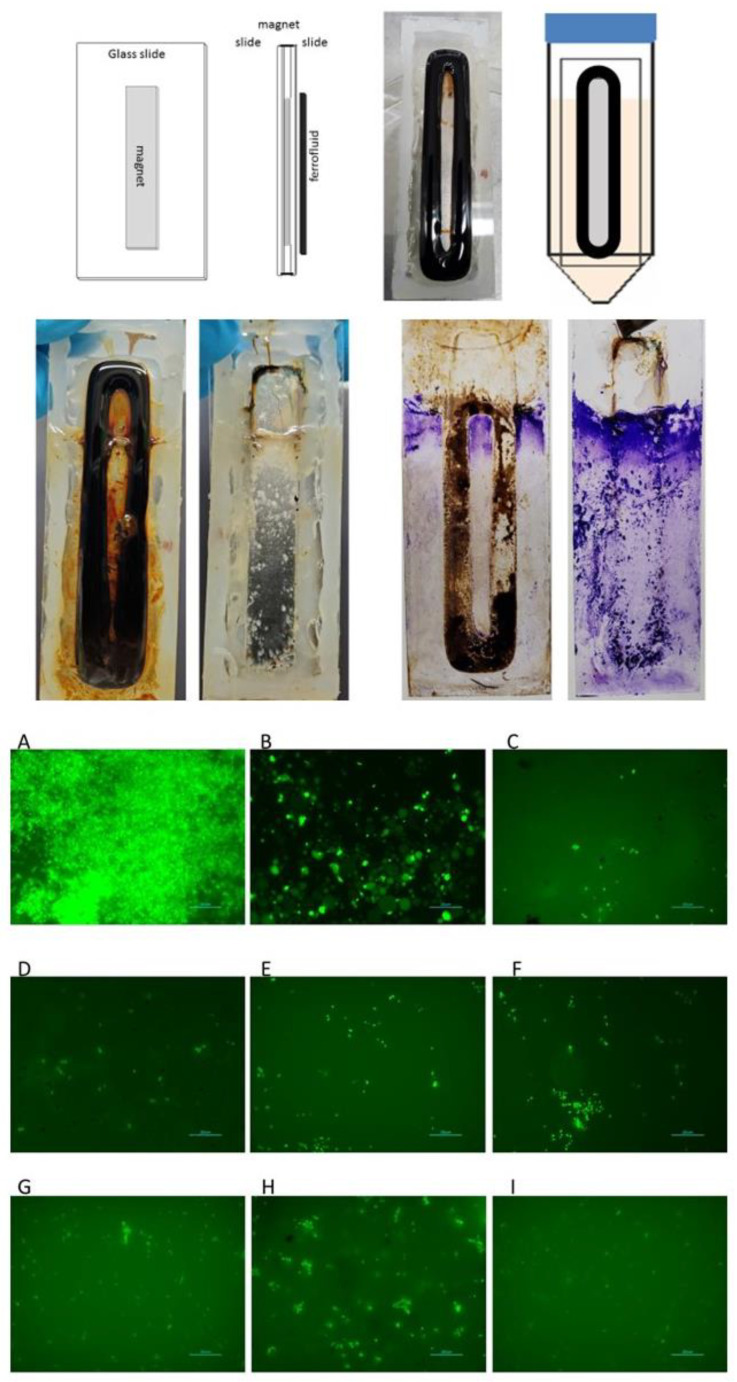
Biofilm formation inhibition by ferrofluid. Upper panel—schematic of slide construction, ferrofluid location, and experimental setup; center panel—biofilm growth on microscope slides (obverse and reverse) not stained (left) and stained with crystal violet; bottom panel—microscopic analyses of *E. coli* slides. (**A**) Control, (**B**) EHF3, (**C**) K5, (**D**) MRF-122EG, (**E**) AMT-Dampro, (**F**) AMT-Magnaflo, (**G**) AMT-Smartec, (**H**) AMT-Rheotec, (**I**) FM. Blue bar scales represent 20 µm.

**Table 1 microorganisms-11-01515-t001:** Properties of SMART fluids.

Group	Ferrofluid	Carrier	Viscosity	Magnetic Particles
1	FerroTec EFH1	Light Hydrocarbon	6 mPa·s	7.8% **
	FerroTec EFH3	Light Hydrocarbon	12 mPa·s	12% **
2	FerroTec PBG200	Water-miscible	<2 mPa·s	4% **
	FerroTec EMG605	Water	<5 mPa·s	4% **
	FerroTec EMG705	Water	<5 mPa·s	4%
3	FerroTec EMG900	Light Hydrocarbon Oil	60.0 mPa·s	17.7% **
	FerroTec EMG901	Light Hydrocarbon Oil	>10.0 mPa·s	11.8% **
	FerroTec EMG905	Light Hydrocarbon Oil	>5.0 mPa·s	7.8% **
	FerroTec EMG909	Light Hydrocarbon Oil	>5.0 mPa·s	3.9% **
	FerroTec EMG911	Light Hydrocarbon Oil	>5.0 mPa·s	2.0% **
4	MRF-122EG	Light Hydrocarbon Oil	42 mPa·s	72%
	AMT-DAMPRO	Light Hydrocarbon Oil	55 mPa·s	76%
	AMT-MAGNAFLO	Light Hydrocarbon Oil	98 mPa·s	80%
	AMT-SMARTEC	Light Hydrocarbon Oil	188 mPa·s	82%
	AMT-RHEOTEC	Light Hydrocarbon Oil	240 mPa·s	84%
5	Kerosene 1.625 (K1)		0 mPa·s *	1.69%
	Kerosene 3.75 (K2)		0 mPa·s *	3.37%
	Kerosene 7.5 (K3)		0 mPa·s *	6.75%
	Kerosene 15 (K4)		0 mPa·s *	13.5%
	Kerosene 30 (K5)		0 mPa·s *	27%
	Kerosene 45 (K6)		0 mPa·s *	40.5%
	Kerosene 60 (K7)		0 mPa·s *	54%
6	Total Home Hand Soap (TH)	Total Home Hand Soap	6635 mPa·s *	27%
	HIGHMARK Dishwashing Liquid (DL)	HIGHMARK Dishwashing Liquid	5010 mPa·s *	27%
	Seventh Generation Inc. Powered by Plants Dish Liquid (PP)	Seventh Generation Inc. Powered by Plants Dish Liquid	3162 mPa·s *	27%
7	Aliver Facemask (FM)		8,6725 mPa·s *	n.d.

* Viscosity measured by NDJ-5S Viscosimeter with S1 or S4 probes at room temperature. 0 mPa·s—means below detection level. ** Percent Concentration is based on calculations from Saturation Magnetization and supported by the manufacturer.

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
