# Peer review of "Using SMART Magnetic Fluids and Gels for Prevention and Destruction of Bacterial Biofilms"

_microorganisms, 2023, doi:10.3390/microorganisms11061515_

Round 1

Reviewer 1 Report

In this manuscript the authors reported the investigation of SMART magnetic fluids and magnetic fields to remove bacterial biofilms from culture tubes, glass slides, multiwell plates, flow cells, and catheters.The ability to remove biofilm increased with the amount of magnetic particles therefore MRFs, FG and homemade FFs with high amount of iron oxide were the most efficient.Some interesting results are obtained. I therefore recommend an acceptance for publishing after next revisions.

1.Pages 2, abstract part, some background sentences can be added;

2.Introduction part, if possible, some important and relative reports about self-assembled gel nanostructures from various styles (Advanced Materials, 2016, 28:3669-3676.; Small, 2020, 16:1907309.; Science China Materials, 2021, 64(4): 942-952.) should be added to show clear background;

3. what about the stability for nanocomposite SMART magnetic fluids, pleas add more describe?

5. Some minor Language error and style should be modified;

In this manuscript the authors reported the investigation of SMART magnetic fluids and magnetic fields to remove bacterial biofilms from culture tubes, glass slides, multiwell plates, flow cells, and catheters.The ability to remove biofilm increased with the amount of magnetic particles therefore MRFs, FG and homemade FFs with high amount of iron oxide were the most efficient.Some interesting results are obtained. I therefore recommend an acceptance for publishing after next revisions.

1.Pages 2, abstract part, some background sentences can be added;

2.Introduction part, if possible, some important and relative reports about self-assembled gel nanostructures from various styles (Advanced Materials, 2016, 28:3669-3676.; Small, 2020, 16:1907309.; Science China Materials, 2021, 64(4): 942-952.) should be added to show clear background;

3. what about the stability for nanocomposite SMART magnetic fluids, pleas add more describe?

5. Some minor Language error and style should be modified;

Author Response

Dear Reviewer,

Thank you very much for this positive review. 

To address your comments we modified the manuscript accordingly.

We modified the abstract a little. Unfortunately, the word limit does not allow us to add more background.

We thank you for the suggestions about self assembled gel nanostructures. It is an interesting topic, however not really related to the manuscript. There are many nanostructures and nanoparticles with biological applications but it is not the scope of this manuscript to present and discuss them all. Although, the   "Small" paper has some interesting data and has been added to the introduction. Thank you. 

What about the stability for nanocomposite SMART magnetic fluids, pleas add more describe?

This manuscript is rather a "proof of concept" method of removing biofilms from different surfaces using some commercially available SMARFs. As we used commercial SMARFs, we did not really focused on their properties and characterization beyond the basic things available through the manufacturers. We leave the details to some specialists who would like to continue this kind of research in the future. 

All small language and format issues have been corrected.

Thank you

Reviewer 2 Report

Overall, correct but still somewhat confusing work. I think that biggest problem is too many of different SMARTF tested, on too many different glassware, and different methods for biofilm removal control. This make reader a little bit confused and hard to deduce differences between different SMARTFs. However, if goal was to confirm that they can remove biofilms to some extent without need to precisely detect differences between different SMARTF then this is OK. Authors  need to write few sentences to explain how/why they decided to use different forms of magnets and different methods for cell detachment control. Other small comments and corrections are given in attached PDF file.

Minor errors are marked as comments in PDF.

Author Response

Dear Reviewer,

Thank you very much for your positive comments. Your impression is absolutely correct, in this manuscript we focused more on the ability of different commercial SMARTFs to remove bacterial biofilms without digging into the process itself. Biofilm is a very broad definition, and many different techniques have been developed to study  bacterial biofilms. To show possible versatile applications we decided to use a few different models.  We added some paragraphs in the methods section including magnets and surface adhesion. We used commercial magnets available on ebay, dimensions of these magnets are determined by the kind of biofilm growing devices: 16mmID for 16mm OD of culture tubes, 6.35mm ID for 6mm OD catheter and so on. The strength of these magnets are determined by their size and material (N52 neodymium).

 Surface adhesion was representing stickiness and tackiness and was simply tested by the drop migration distance and trace patterns. It is very rough and simple descriptive method as we do not have any equipment to measure it. As the entire manuscript is just a proof of concept we thought that it is enough to notice these differences between SMARTFs rather than ignore them completely.

Why glass slides were not "slided" also 12 times? 

It is very hard to control FFs on the flat open surface, when we tried to clan the edges of the slides we were loosing some volume and it was not possible to move the FFs "12 times"

Saturation Magnetization data was provided by manufacturers in their fliers. we stated that 0 viscosity is below detection level ( Thank you for catching this)

How the used amount of washing magnetic fluid was chosen and how the magnet shape was chosen? Was it the according the surface of the dish, its volume,?

The amount of FF was chosen arbitrary based on the size of  biofilm device and magnet. In some cases it does not really matter (slides or flow cells), but in some like catheters a higher volume will block the entire duct. 

All other small errors have been corrected it the text.

Thank you

Reviewer 3 Report

This is an interesting topic, but the paper is very poorly written.  The manuscript doesn't follow the format requirements of this journal.

1. The abstract should highlight the novelty of the work.

2. The authors names in citations doesn't need to be in all caps.

3. The authors of this work, failed to include references of the most notable works in ferrofluids and magnetophoresis.  The paper needs to be revised to include the following:

Hewlin, R.L., Jr., Edwards, M., and Schultz, C., "Design and Development of a Traveling Wave Ferro-microfluidic Device and System Rig for Potential Magnetophoretic Cell Separation and Sorting in a Water-Based Ferrofluid", Micromachines 2023, 14, 889.

Yellen, B.B.; Hovorka, O.; Friedman, G. Arranging matter by magnetic nanoparticle assemblers. Proc. Natl. Acad. Sci. USA 2005, 102, 8860–8864. 

Yellen, B.B.; Erb, R.M.; Son, H.S.; et al., Travelling Wave Magnetophoresis for High Resolution Chip Based Sepa-rations. Lab Chip 2007, 7, 1681–1688. 

4. The figures need to be placed in the body of the manuscript.

5. The main contributions of this work should be bullet pointed in the end of the literature review discussion.

6. The methods and materials section is vague, there needs to be more in depth process steps included.  This can also be included in supplementary materials.

7. The quality of the figures are poor and needs to be addressed.

8. There is no conclusion section.

The paper is very poorly written and needs to be revised. 

Author Response

Dear Reviewer,

Thank you very much for this positive review. 

To address your comments we modified the manuscript accordingly.

  • The manuscript has been edited by two native speakers to improve the language.
  • I dont understand the format requirements comment. Journal requirements include: Introduction, Materials and Methods, Results, Discussion, Conclusions (optional). We added a separate conclusions chapter to summarize the article.
  • The citations are added automatically from the Endnote. I believe it will be fixed in the final version.
  • We added the recent citation,  Hewlin, R.L., Jr., Edwards, M., and Schultz, C., "Design and Development of a Traveling Wave Ferro-microfluidic Device and System Rig for Potential Magnetophoretic Cell Separation and Sorting in a Water-Based Ferrofluid", Micromachines 2023, 14, 889.
  • Figures have been automatically formatted and inserted in the article body. We have no control on this process but I think it will be corrected in the final version.
  • We added some conclusions to the manuscript.
  • All figures are high resolution .jpg files with good quality. Converting to PDF is undoubtedly the problem here. It will be fixed in the final editorial steps. 

Thank you

Round 2

Reviewer 2 Report

Now it is ok.

Reviewer 3 Report

Please address the formatting 

n/a